# ICAMs in Immunity, Intercellular Adhesion and Communication

**DOI:** 10.3390/cells13040339

**Published:** 2024-02-14

**Authors:** Claudia Guerra-Espinosa, María Jiménez-Fernández, Francisco Sánchez-Madrid, Juan M. Serrador

**Affiliations:** 1Immune System Development and Function Unit, Centro de Biología Molecular “Severo Ochoa”, Consejo Superior de Investigaciones Científicas-Universidad Autónoma de Madrid, 28049 Madrid, Spain; cguerra@cbm.csic.es; 2Immunology Department, Instituto de Investigación Sanitaria Hospital Universitario La Princesa, Universidad Autónoma de Madrid, 28006 Madrid, Spain; jmzfzmaria@gmail.com (M.J.-F.); fsmadrid@salud.madrid.org (F.S.-M.); 3Vascular Pathophysiology Area, Centro Nacional de Investigaciones Cardiovasculares (CNIC), 29029 Madrid, Spain; 4CIBER de Enfermedades Cardiovasculares (CIBERCV), Instituto de Salud Carlos III, 28029 Madrid, Spain

**Keywords:** ICAMs, β2-integrins, moesin, leukocytes, intercellular adhesion

## Abstract

Interactions among leukocytes and leukocytes with immune-associated auxiliary cells represent an essential feature of the immune response that requires the involvement of cell adhesion molecules (CAMs). In the immune system, CAMs include a wide range of members pertaining to different structural and functional families involved in cell development, activation, differentiation and migration. Among them, β_2_ integrins (LFA-1, Mac-1, p150,95 and α_D_β_2_) are predominantly involved in homotypic and heterotypic leukocyte adhesion. β2 integrins bind to intercellular (I)CAMs, actin cytoskeleton-linked receptors belonging to immunoglobulin superfamily (IgSF)-CAMs expressed by leukocytes and vascular endothelial cells, enabling leukocyte activation and transendothelial migration. β2 integrins have long been viewed as the most important ICAMs partners, propagating intracellular signalling from β2 integrin-ICAM adhesion receptor interaction. In this review, we present previous evidence from pioneering studies and more recent findings supporting an important role for ICAMs in signal transduction. We also discuss the contribution of immune ICAMs (ICAM-1, -2, and -3) to reciprocal cell signalling and function in processes in which β2 integrins supposedly take the lead, paying particular attention to T cell activation, differentiation and migration.

## 1. Introduction

The establishment of interactions among leukocytes and leukocytes with immune-associated auxiliary cells, such as endothelial cells in blood and lymphatic vessels, and epithelial cells in skin, lungs and the gastrointestinal tract, is essential to arm immune responses [1,2]. Homotypic and heterotypic cell adhesions between different cell types require the involvement of cell adhesion molecules (CAMs), which are proteins expressed in the plasma membrane (PM) and held at intercellular contacts by cytoskeleton-dependent forces [3]. In the immune system, CAMs include a wide range of members, most of them belonging to cadherins, claudins, occludins, selectins, integrins and the immunoglobulin superfamily (IgSF). These receptors regulate cell development, activation, differentiation, migration and many other cellular processes of crucial importance for the immune response. Cadherins are components of the adherens junctions that stabilize cell-cell adhesion between vascular endothelial (VE-cadherin) or epithelial (E-cadherin) cells in a Ca^2+^-dependent manner. Claudins and occludins are extracellular two loop-containing tetraspan proteins found in endothelial tight junctions. Cadherins, claudins and occludins form homophilic interactions at intercellular junctions that are transiently disrupted during leukocyte diapedesis to lead the immune response to the anatomical domains where it is required [4]. 

Selectins are a family of transmembrane glycoproteins that share homology with Ca^2+^-dependent C-type lectins and are expressed on leukocytes (L-selectin), platelets (P-selectin) and endothelial cells (P- and E-selectin) [5]. Selectins, together with their ligand P-selectin glycoprotein ligand 1 (PSGL-1) play a predominant function in leukocyte homing to lymph nodes (LNs) and the skin, enabling leukocyte tethering and rolling on endothelial cells, the two initial steps of transendothelial migration (TEM) [6]. PSGL-1 and L-selectin establish heterophilic interactions with their counter-receptors E/P-selectin and the mucin CD34, respectively, on activated endothelial cells of postcapillary venules, and L-selectin binds to sialyl Lewis X (sLe^x^) carbohydrates on MadCAM-1 and GlyCAM-1 expressed by high endothelial venules (HEVs) of lymphatic vessels [7,8]. Ligand-induced selectin signalling, together with the binding of chemokines to their receptors, triggers the transactivation of integrins, which promotes the transition from leukocyte rolling to firm adhesion [9,10]. 

Integrins are α/β heterodimeric receptors for extracellular proteins of the matrix, such as collagen, laminin, fibrinogen, fibronectin and vitronectin, but also for intercellular adhesion counter receptors [11]. So far, eight different β-integrin subunits have been described, of which β1, β2 and β7 are expressed in leukocytes. Among them, β2 integrins (LFA-1, Mac-1, p150,95 and α_D_β_2_) are involved in both homotypic and heterotypic leukocyte adhesion [12]. LFA-1 (α_L_β_2_, CD11a/CD18) is the major β_2_ integrin expressed on lymphocytes, but also monocytes and neutrophils can express it [13,14], whereas Mac-1 (α_M_β_2_, CD11b/CD18) is characteristic of myeloid cells, and p150,95 (α_X_β_2_, CD11c/CD18) and α_D_β_2_ (CD11d/CD18) are primarily expressed on monocytes/macrophages and monocyte-derived dendritic cells [15,16,17,18]. Integrins are flexible molecules, whose ligand binding is regulated by both conformational changes in their backbone (i.e., affinity) and the clustering of receptors (i.e., avidity) [19]. Integrin molecular interactions with their ligands take place at the metal ion-dependent adhesion site (MIDAS) of the αI domain, which provides three of the coordinate covalent bonds with divalent cations (Mg^2+^/Mn^2+^); the fourth bond is provided by a glutamate/aspartate in the ligand [20,21]. Divalent cations and ligand binding shift the shape of integrins from closed/inactive to open/active conformational states [22,23]. In addition to conformational changes mediated by outside-in signals, inside-out mechanotransduction can change integrins from a bent shape in the inactive closed conformation of the I domain to an active open conformation. This is achieved by mechanical forces generated by F-actin polymerization and contractility upon its interaction with the actin-binding proteins Kindlin-3 and Talin, connecting F-actin to the cytoplasmic tail of the integrin β chain [24,25,26]. It is thought that these forces can pull the β-chain cytoplasmic tail away from the α chain, promoting the exposition of the ligand-binding site in the I domain of the integrin head. 

β2 integrins bind intercellular adhesion molecules (ICAMs), receptors belonging to the IgSF-CAMs that are expressed by leukocytes and vascular endothelial cells and take part in leukocyte activation and TEM [27,28,29]. IgSF-CAMs are cell-surface glycoproteins, whose main characteristic is the expression of Ca^2+^-independent cell–cell adhesion immunoglobulin-like loops in their extracellular domain [30,31]. Nectins, platelet endothelial CAMs (PECAMs) and junctional adhesion molecules (JAMs) in endothelial cell–cell junctions, mucins (e.g., MadCAM-1 and GlyCAM-1) and vascular CAM (VCAM-1) in endothelial cells and CD2, the CD2 counter-receptor CD58 (LFA-3), the CD2-related signalling lymphocyte activation molecules (SLAMs) and ICAMs in leukocytes are some of the many CAMs that belong to the IgSF and play an important role in the immune system. Of these, only ICAMs (ICAM-1 to -5) can establish intercellular interactions with β2 integrins. ICAM-1, -2 and -3 are expressed by both leukocytes and endothelial cells, although only bone marrow and tumour endothelial cells can express ICAM-3 [32,33]. ICAM-4 (Landsteiner–Wiener blood group) is restricted to erythrocytes and erythroid precursors and mediates sickle red cell adhesion to endothelial cells via the integrins p150,95 and αvβ3 [34,35]. ICAM-5 (telencephalin) binds to LFA-1 and is specifically expressed in neuronal filopodia, stimulating neurite outgrowth and dendrogenesis on presynaptic cells and promoting the maturation of synaptic contacts by binding to VLA-5 [36]. Upon the β2 integrin–ICAM adhesion receptor interaction, integrins rather than ICAMs have long been considered as the main signalling molecules [37]. This role has been ascribed to their association with the major intracellular signalling networks driving cell communication. In this review, we will discuss previous evidence and more recent findings that also support an important role for ICAMs in triggering intracellular signalling. The aim of this review will not be to address the large number of signalling pathways triggered by different CAMs in immune and immune-associated cell types, as this has been exhaustively reviewed elsewhere [38,39], but to discuss how immune ICAMs (ICAM-1, -2, and -3) contribute to reciprocal cell signalling and function in processes in which β2 integrins are thought to take the lead, with particular emphasis on T-cell activation, differentiation and migration.

## 2. ICAMs in the Immune System

ICAMs share a considerable homology, although human ICAM-1 and -3 are located on the short arm of chromosome 19, and ICAM-2 is located on chromosome 17, suggesting that all three molecules evolved from a common primordial ICAM-coding gene [40]. ICAM-1 and -3 both have five C2-type Ig-like domains (D1-5) and share 52% of their amino acid sequence (38% D1 and 77% D2). ICAM-2, meanwhile, has only two domains (D1-2), but these are very closely related to D1 and D2 of ICAM-3 [41,42]. On the other hand, the cytoplasmic tails of ICAM-2 and -3 are much less similar. ICAM-3 contains more amino acid residues susceptible to phosphorylation than its homologues, a feature that has led to a widespread expectation about the specific functions that this region might play in signalling.

A major feature of ICAMs is their high level of glycosylation (Figure 1), which is necessary for translocation to the PM and ligand binding [43,44]. While protein glycosylation takes place on Asn (N-glycosylation) and Ser/Thr (O-glycosylation), ICAMs are mainly N-glycosylated [45]. Eight potential N-glycosylation sites distributed over D2-4 are found on ICAM-1, which have given rise to different glycosylated forms depending on the cell type and function [45,46,47]. For example, the high-mannose form of ICAM-1 was found to be more efficient in regulating monocyte rolling and adhesion but shows altered interaction with ERM proteins, causing a reduction in the lateral dynamics of ICAM-1 clustering [48], and the complex N-glycan form of ICAM-1 was required for cytoskeletal changes in endothelial cells, affecting vascular permeability [49]. On the other hand, ICAM-3 has fifteen potential N-glycosylation sites distributed over all domains, although only half of these sites are actually N-glycosylated [50]. ICAM-3 is, thus, the most glycosylated ICAM compared to the eight and six N-glycosylation sites of ICAM-1 and -2, respectively [51]. It has been suggested that glycan residues on ICAMs may influence dimerization. ICAM-1 can dimerize through D1-D1 interactions, whereas ICAM-3 lacks this ability [52]. The absence of glycosylation on D1 of ICAM-1 and its presence on the same domain of ICAM-3 would provide a plausible explanation for this structural difference and also for the selection of ICAM-1 as a pathogen receptor, since pathogens bind ICAM-1 through D1. High-mannose carbohydrates specifically present on the D2 of ICAM-2 and ICAM-3, but not on the D2 of ICAM-1, could also confer selectivity for DC-specific ICAM-3-grabbing non-integrin (DC-SIGN) binding to the D2 of ICAM-2 and -3 [53].

### 2.1. ICAM-1 (CD54)

ICAM-1 is by far the most studied ICAM, not only due to its inducible expression by endothelial cells, leukocytes and many other cell types, which makes it a compelling cellular marker of activation and inflammation, but also due to its essential function in endothelial cells as counter-receptors for LFA-1 in lymphocytes and Mac-1 in myeloid cells (Figure 1) during leukocyte migration [54]. ICAM-1 is also an important receptor for human infectious pathogens, such as common cold rhinoviruses and coxsackieviruses [55,56,57]—bound to ICAM-1 by a canyon-shaped depression formed around the fold vertices of the viral icosahedral capsid,—or *Plasmodium falciparum*-infected erythrocytes [58,59] and *Toxoplasma gondii* [60],—which bind ICAM-1 through the adhesin MIC2 and the protein PfEMP1, respectively,—facilitating viral entry, and parasite tissue invasion through endothelial and epithelial tissue barriers. 

Mature ICAM-1 is a glycoprotein of 76–114 kDa that forms homodimers through D1 contacts, and also W-shaped tetramers organized into a band-like one-dimensional cluster of D1-D1 contacts connecting ICAM-1 dimers through D4-D4 contacts [52,61], a molecular structure that may increase ICAM-1 affinity for β2 integrins. Recent super-resolution microscopy studies have shown that β2 integrins may bind ICAM-1 dimers expressed on the same cell in cis [62]. In addition to their extended high-affinity structure that binds ICAMs expressed on an opposite cell in trans, β2 integrins can adopt a bend-shaped active conformation that binds ICAM-1 dimers face-to-face, forming nanoclusters. Functionally, this interaction would prevent β2 integrin binding to ICAM-1 in trans, thus limiting both firm adhesion of immune cells to vessels and their recruitment to inflammatory foci [63]. Further investigations would be required to better understand the role played by cis interactions between β2 integrins and ICAMs in leukocyte migration and activation.

In addition to the full-length protein, six PM-associated alternative spliced isoforms of ICAM-1 have been detected in endothelial cells: ICAM-1 containing two (one variant), three (two variants) and four (two variants) Ig-like domains, with D1 and D5 always present and a splice variant lacking D5 [64]. Whether ICAM-1 isoforms can alter ligand binding and function has not been clearly demonstrated, although studies performed with ICAM-1 mutant mice suggest that the ICAM-1 splice variant containing D1-3 and D5 potentially contributes to the regulation of intracellular proinflammatory signalling events in an experimental model of autoimmune encephalomyelitis [65,66].

Upon receptor aggregation, ICAM-1 can be phosphorylated on Tyr485 by Src and Met tyrosine-protein kinases [67,68]. This posttranslational modification allows type-1 matrix metalloproteinases (MT1-MMP) to interact with ICAM-1 in endothelial cells and to cleave the ICAM-1 ectodomain, resulting in soluble (s)ICAM-1, a process that seems important for leukocyte TEM and cancer metastasis [69]. sICAM-1 is also generated under pathological conditions by both human leukocyte elastase (HLE)-mediated proteolytic cleavage of ICAM-1 and the translation of an alternative mRNA transcript [70,71], interfering with Mac-1/ICAM-1 interactions and neutrophil activation and extravasation. Elevated serum levels of sICAM-1 have been reported in some inflammatory conditions, such as LAD [72], cirrhosis associated with COVID-19 or bacterial sepsis [73], cancer [74], systemic lupus erythematosus [75], psoriasis [76] or rheumatoid arthritis [77], suggesting sICAM-1 as an inflammatory marker. 

However, the most important regulation of ICAM-1 takes place at the transcriptional level. ICAM-1 increases on the PM in response to pathogen-derived pro-activatory stimuli (e.g., binding of antigen (Ag) to the T-cell receptor (TCR) in T lymphocytes and binding of lipopolysaccharides (LPS) to CD14 and TLR4 in monocytes, neutrophils and endothelial cells) [78,79], growth factors (e.g., vascular endothelial growth factor (VEGF) in endothelial cells) [80], pro-inflammatory cytokines (e.g., TNF-α, IFN-γ and IL-1β in both leukocytes and endothelial cells) [81] and reactive oxygen species (ROS) [82]. Transcription of the ICAM-1-coding gene (*ICAM-1*) is mainly controlled by the proximal gene promoter region, which contains functional binding sites for activator protein 1 (AP-1), specificity protein 1 (SP1), nuclear factor-kappa B (NF-κB), interferon regulatory factors (IRF), ETS1 and signal transducer and activator of transcription (STAT)-1 [83]. Phosphorylation of phosphoinositide-3-kinase (PI3K)-Akt, mitogen-associated protein kinases (MAPKs), jun kinases (JNKs), Janus tyrosine kinases (JAKs) or the cytoskeleton-associated focal adhesion kinase (FAK) and proline-rich tyrosine kinase 2 (Pyk2) are among the major signalling pathways involved in *ICAM-1* transcriptional activation [84]. Of particular interest is the PI3K/Akt pathway induced by growth factors such as VEGF, which increases ICAM-1 expression in microvascular endothelial cells through the phosphorylation of eNOS on Ser1177 and nitric oxide (NO) production [85]. Remarkably, studies on ICAM-1-deficient mice demonstrated that the combined action of VEGF and ICAM-1 is required to induce ICAM-1 expression, as VEGF-dependent chemotaxis, eNOS phosphorylation and NO production were all impaired in endothelial cells from ICAM-1-deficient mice [86,87]. These findings would be in accordance with more recent reports, showing that ICAM-1 regulates eNOS activation/phosphorylation and NO production in endothelial cells, leading to increased PECAM-1 adhesion, phosphorylation-mediated VE-cadherin disorganization at adherens junctions and actin cytoskeleton rearrangements that promote cell motility and TEM [88,89]. 

ICAM-1 expression is also characteristic of tumours. ICAM-1 on melanoma, breast, lung and colorectal cancer cells is associated with malignancy [68,90,91,92] and involved in tumour initiation, progression and metastasis. This aggressive phenotype of circulating cancer cells is not only conferred by facilitating TEM from the endothelial side but also by enabling collective migration of cell aggregates between ICAM-1 on cancer cells and β2 integrins on leukocytes, which can promote the proliferation of secondary tumours trapped in post-capillary venules [93]. In this regard, recent studies have shown that posttranslational N-myristoylation of ICAM-1 on D1 (Gly23) increases adhesion but reduces the migration of cancer cells [94]. This modification is mediated by N-myristoyltransferase-1 (NMT-1) and interferes with the recognition of ICAM-1 by FBXO4 in the ubiquitin E3 ligase Skp-1-Cullin-F-box (SCF) complex, thereby reducing proteasome-mediated degradation of ICAM-1. Whether NMT-1 can prolong ICAM-1-mediated adhesion for longer in secondary tumours remains to be investigated.

ICAM-1 is also present on leukocyte- and tumour-cell-derived exosomes [95,96], small double-membrane extracellular vesicles formed within late endosomal compartments. ICAM-1-LFA-1 binding between tumour-cell-derived exosomes and leukocytes can interfere with leukocyte adhesion to activated endothelial cells and their recruitment to tumours [96] but also with the activation of cytotoxic CD8^+^ T cells [97], whose interaction with IFN-γ-induced exosomes containing the checkpoint molecule PD-L1 seems to be one of the numerous mechanisms for immune evasion by cancer cells.

### 2.2. ICAM-2 (CD102)

ICAM-2, the less well-characterized immune ICAM, is a glycoprotein of approximately 55–80 kDa, whose extracellular region contains two Ig domains (D1-2) (Figure 1). ICAM-2 was identified by the inability of blocking anti-ICAM-1 Abs to completely inhibit leukocyte adhesion to endothelial cells in vitro [41]. ICAM-2 is constitutively expressed on lymphocytes, monocytes and endothelial cells, but is almost absent on granulocytes. Unlike ICAM-1, a fraction of ICAM-2 is localized near endothelial junctions and promotes angiogenesis through homotypic adhesion, Rac-1 signalling, increased cell migration and tubule formation, but it does not stimulate c-*fos* transcription, activate RhoA or alter actin cytoskeleton organization in endothelial cells [98,99] (Figure 2A). Like ICAM-1, ICAM-2 is of great importance for the establishment of leukocyte-endothelial interactions. ICAM-2 facilitates neutrophil crawling and TEM through D1 binding to Mac-1 in a cytokine-dependent manner [100] and chemokine-induced migration of dendritic cells (DCs) across endothelial cells through D2 binding to DC-SIGN [101], a type II transmembrane mannose-binding C-type lectin mostly expressed on macrophages and monocyte-derived DCs. NK cell migration and NK cell-mediated clearance of tumour and pathogen-infected cells are also important processes in which ICAM-2 has been involved as a counter-receptor of LFA-1 [102,103].

### 2.3. ICAM-3 (CD50)

ICAM-3 [42,51,104] is a highly glycosylated protein of 110–160 kDa encoded by a gene (*ICAM-3*) phylogenetically derived from *ICAM-1* by gene duplication in humans [40]. ICAM-3 is constitutively expressed on all leukocytes and can be particularly present on some endothelial and tumour cells (e.g., 133^+^ non-adherent endothelial-forming cells and lymphomas) [32,105] and regulated at the transcriptional level by the transcription factor RUNX3 during the transendothelial migration of monocytes and their differentiation to macrophages [106]. ICAM-3 expression on leukocytes is also regulated by activation-induced shedding mechanisms, resulting in sICAM-3 under pathophysiological conditions [107]. LFA-1, α_D_β_2_ and DC-SIGN have been identified as the major counter receptors of ICAM-3 [16,17,108,109] (Figure 1); however, a large number of studies have shown that the affinity of ICAM-3 for LFA-1 is approximately half that for αDβ_2_ and nine-times lower than the affinity of ICAM-1 for LFA-1 [16,110,111,112]. Because ICAM-1 is almost absent on resting T cells and their ICAM-2 levels are very low compared to ICAM-3, a preponderant role has been assigned to ICAM-3/LFA-1 and ICAM-3/α_D_β_2_ interactions in the early stages of the adaptive immune response, when naïve T lymphocytes establish first contacts with Ag presented by antigen-presenting cells (APCs) [113,114,115,116]. In this context, previous studies have shown that DC-SIGN binds ICAM-3 with higher affinity than LFA-1 or α_D_β_2_, and although monocyte-derived DCs express both LFA-1 and DC-SIGN, the latter is currently considered a better ICAM-3 partner to trigger costimulatory signals within LNs during cognate T cell–APC interactions [109]. On the other hand, unlike ICAM-1 and -2, ICAM-3 is absent in rodents, whose genome has probably lost it through gene deletion during the evolution of mammals [40]. This condition has long been a major limitation for the study of the pathophysiological functions of ICAM-3. In the absence of ICAM-3-engineered mice, which could help to explore the function of ICAM-3 in a whole organism, all efforts to clarify whether its physiological functions may be overcome by ICAM-1 and ICAM-2 or are unique and irreplaceable have been rather inconclusive. On the other hand, this singularity of ICAM-3 could also be seen as an opportunity to study cell adhesion mechanisms specifically involved in human immunity. 

## 3. ICAMs Anchor to the Actin Cortex

ICAMs are dynamic adhesion molecules intimately associated with the actin cortex of the PM; thus, their functions should be understood in the context of their linkage to the cytoskeleton. However, despite the fact that their relationships have been extensively studied over the last few decades, how the cytoskeleton regulates the function of ICAMs is still much debated. The association of ICAMs with the actin cytoskeleton is mediated by their interaction with actin-binding proteins (ABPs), which not only provides stability at intercellular contacts but also helps to generate intracellular signals. Although the interaction of ICAM-1 and -2 with α-actinin has been described [117,118], the most studied ABPs involved in ICAM signalling and anchoring to the actin cytoskeleton are the ERMs [119]. ERM stands for ezrin, radixin and moesin, a family of proteins that act as key regulators for the formation of the uropod, a PM protrusion for adhesion, migration and signalling at the rear of motile leukocytes, wherein ICAMs are concentrated [120]. ERMs link the cytoplasmic tails of transmembrane proteins to actin filaments and organize them underneath the PM. Leukocytes mainly express ezrin and moesin, with moesin being more abundant than ezrin. ERMs interact with ICAM-1, -2 and -3 in a phosphatidylinositol 4,5-bisphosphate (PIP2)-dependent manner at the uropods of leukocytes [121,122,123], where ICAMs are redistributed in response to chemotactic factors (e.g., chemokines and FMLP) [124,125,126]. The polarization of ICAMs at the uropods of leukocytes facilitates intercellular contacts, for instance, during the recognition of target cells by NK cells [127], the recruitment of bystander leukocytes during TEM [128] or the final step of leukocyte extravasation, in which LFA-1-mediated adhesion to the subendothelial layer delays the detachment of the uropod to co-ordinately regulate the recruitment of leukocytes to inflamed tissues [129]. ERMs can bind directly to ICAM-1, -2 and -3 via a positively charged juxtamembrane cytoplasmic region bearing a contiguous nonpolar amino acid motif [122,130]. In addition to this consensus motif, specific serines within the cytoplasmic tail of ICAMs may regulate their binding to ERMs through phosphorylation-dependent mechanisms. In support of this, serines of the ICAM-3 cytoplasmic tail, some of them susceptible to phosphorylation by PKC-θ, interact with the N-terminal 4.1(F)ERM domain of ERMs, and phosphomimetic mutations of these residues interfere with their binding to the FERM domain, most likely by reducing the net positive charge of their FERM-binding motifs [122,131]. Activated ERMs and ICAMs co-operate together in the organization of microvilli, F-actin-based finger-shaped PM protrusions that are important for TCR signalling and leukocyte tethering/rolling during the initial contacts with APCs and endothelial cells, respectively [121,132,133,134]. ERMs organize ICAM-1 on the apical side of endothelial cells, enabling leukocyte crawling and subsequent firm adhesion and arrest. In response to the interaction of β2 integrins with ICAM-1, endothelial cells form an F-actin-based docking structure, which embraces leukocytes with ICAM-1-rich PM processes containing PIP2 and phosphorylated moesin and ezrin [135,136]. This actin cytoskeleton-based structure is thought to allow for the dynamic transition between leukocyte firm adhesion and TEM. In addition, the association of ICAM-1 with the actin cytoskeleton and its translocation into caveolae has been involved in lymphocyte transcellular migration [137], although the precise mechanisms by which this process takes place remain to be investigated. 

The dynamic anchoring of ICAMs to the actin cytoskeleton also regulates the interactions between T cells and APCs. It has been proposed that the reduced mobility of ICAM-1 on APCs provides resistance to the opposing forces of LFA-1 on T cells and facilitates ligand-dependent LFA-1 activation at the immune synapse (IS) [138], a PM-associated intercellular structure that regulates adhesion and signalling during cognate interactions between T cells and APCs. At the IS, the mobility and clustering of ICAMs are regulated by changes in the expression and activation of ERMs and other ABPs (e.g., α-actinin), which organize LFA-1/ICAMs into a peripheral supramolecular activation cluster (p-SMAC). At this compartment, the regulation of the extent of LFA-1/ICAM interaction by cytoskeletal forces can control TCR-triggered activation [139]. 

## 4. ICAMs as Bidirectional Signalling Receptors

In addition to the signalling triggered by the binding of peptide (p)-Ag-major histocompatibility complex (MHC) to the TCR, full activation, differentiation and effector functions of T cells require costimulatory signals provided by accessory molecules [140]. Although the best characterized costimulatory molecules on T cells are members of the CD28 family (i.e., CD28/ICOS as activators and CTLA-4/PD-1 as suppressors), a second signal is also delivered from the interaction between ICAMs and their ligands at both sides of the IS.

### 4.1. LFA-1 Signalling by ICAM Binding

In addition to enhancing intercellular adhesion, ICAMs on APCs can deliver costimulatory signals via LFA-1. Supporting this, pioneering in vitro studies using soluble recombinant ICAMs, as well as ICAM-transfected L cells expressing allogenic MHC-II molecules, showed that binding of ICAM-1, -2 and -3 to LFA-1 on T cells synergized with costimulatory signalling from CD28, promoting increased expression of the activation markers CD69 and CD25, ERK phosphorylation and cell proliferation even though preferential production of IL-4 and IL-5 (Th2 cytokines), is via CD28, and IFN-γ and IL-2 (Th1 cytokines) via binding of ICAM-1, -2 and -3 to LFA-1 [141,142,143]. Thus, LFA-1 on T cells delivers not only CD28-shared signals but also specific signalling that determines the production of cytokines that define Th1 and Th2 cell subsets. The Th1 cell program promoted by LFA-1/ICAMs can occur through the inhibition of GATA3, a transcription factor essential for Th2 differentiation, but also by activation of GSK-3β-Notch-1 and the intracellular complement system C3b-CD46, which are γ-secretase-dependent signalling mechanisms that activate Th1-specific transcription factors (e.g., T-bet) and glycolytic metabolic reprogramming [144,145,146,147]. Likewise, ICAM-1 binding to LFA-1 promotes ERK phosphorylation and Ca2^+^ intracellular fluxes, increasing IFN-γ production by subsets of innate T lymphocytes, such as invariant NK, mucosa-associated invariant (MAI) and γδ T cells [148]. In B cells, LFA-1 is also important for activation and differentiation as blocking LFA-1 Abs disrupts B-cell proliferation and Ig production in vitro and during the interaction of resting B cells with β2-deficient helper T lymphocytes [149]. These findings have been confirmed by studying the interaction of LFA-1 on B cells with ICAM-1 embedded on IS-reconstituted planar lipid bilayers, which showed that LFA1/ICAM-1 reduced the Ag levels required for B-cell receptor (BCR)-triggered activation [150]. However, less importance has been given to LFA-1 on DCs at the initial phases of the immune response since some studies have shown that LFA-1 on mature DCs remains inactivated by cytohesin-1 [151], an actin cytoskeleton regulator that binds to CD18 and allows DCs to actively control antigen-driven T-cell proliferation. In this context, there is some controversy about how ICAMs on DCs can regulate T-cell responses. Whereas some studies performed with ICAM-1-deficient mice or ICAM-1-expressing DC-derived exosomes have clearly shown that LFA-1 on either CD4^+^ or CD8^+^ T cells and ICAM-1 on DCs are both required for long-lasting cognate T cell–DC interactions and effective T-cell survival and memory [152,153,154], more recent findings reported that productive short-lived interactions between naïve T cells and cognate DCs at early stages of the immune response seem independent of LFA-1 activation [155,156]. These studies suggested that despite strengthening the IS, the interaction between LFA-1 on naïve T cells and ICAM-1 on DCs is not essential for CD4^+^ and CD8^+^ T-cell effector functions in vivo, at least upon their respective type 1 helper T cell (Th1)- and type 1 cytotoxic T-cell (Tc1)-polarizing conditions, leaving the door open for a role of the interaction between ICAM-1 and LFA-1 in DC-mediated priming and function of other T-cell subsets. In this regard, it has been reported that ICAM-1 but not ICAM-2 is required for the activation, proliferation and differentiation of naïve CD4^+^ T-cells stimulated with myelin-derived peptides presented by splenic DCs in a mouse model of experimental autoimmune encephalitis (EAE) [157], an animal model of multiple sclerosis whose pathogenesis is dependent on Th1 and Th17 effector functions. In this model, ICAM-1- but not ICAM-2-deficient DCs poorly prime naïve CD4+ T cells, although ICAM-1 and -2 on endothelial cells are both required for Th1 and Th17 migration across the blood–brain barrier during the effector phase of EAE. ICAM-1 on DCs is also an important factor for regulatory T cells (Tregs). Blockade or deficiency of LFA-1 on Tregs reduces the expression of CD80/CD86 on DCs and their ability to present Ag to naïve T cells, leading to immune tolerance [158,159]. Interestingly, it has also been reported that ICAM-1 on inflamed dermal vessels is key for Treg homing and function in the skin [160], providing a clear example of how the same ICAM expressed by different cell types can exert LFA-1-mediated complementary actions of importance for the proper function of specific T-cell subsets.

### 4.2. Reciprocal Signalling by ICAMs

In addition to the induction of LFA-1-mediated signalling, the use of mAbs to mimic the binding of LFA-1 to D1 of ICAMs has provided evidence of the contribution of ICAM-1, -2 and -3 to leukocyte activation. In B cells, clustering of ICAM-1 upregulated MHC-II molecules and activated ERK1/2 and the Src kinase Lyn [161], whereas in naïve CD4^+^ T cells, the combination of anti-CD3 plus anti-ICAM-1 or anti-ICAM-3 Abs caused prolonged proliferation in a manner similar to costimulation by LFA-1 [162,163,164,165]. However, costimulation with ICAM-1 provided greater protection against apoptosis and memory T-cell fate, as did CD28 through the activation of different downstream factors since CD28 activated Dtk and FGFR1 receptor tyrosine kinases, whereas ICAM-1 rather activated IGF-1R and HGFR [166,167]. This function of ICAM-1 and ICAM-3 may be complemented by their involvement in the clearance of apoptotic cells, the former on phagocytic macrophages and the latter on apoptotic leukocytes and leukocyte-derived microparticles that stimulate the chemotaxis of phagocytes towards apoptotic immune cells and tissues during the resolution of inflammatory processes [168,169,170]. 

The involvement of ICAMs in apoptosis is supported by the costimulatory ability of ICAM-1 to upregulate the anti-apoptotic protein Bcl-2 but also by the stimulation of the pro-survival PI3K/Akt signalling pathway. Activation of PI3K/Akt leading to inhibition of apoptosis has also been described for ICAM-2 and ICAM-3 in the protection of T cells and B cells against TNF-α and Fas-mediated cell death and in the enhancement of cancer cell survival, migration and invasion [171,172,173], which has been proposed to be dependent on their binding to activated ERM proteins. However, a direct role for ICAM-2 in T-cell costimulation has not been clearly demonstrated. In contrast, ICAM-3 is thought to play an important costimulatory function in LNs during the antigen priming of naïve T lymphocytes by mature DCs. ICAM-3 on T cells interacts with either DC-SIGN or LFA-1 on DCs and transmits intracellular signals that facilitate the activation of LFA-1 and binding to ICAM-1 on DCs as well as MHC-II clustering for a more efficient presentation of antigenic peptides [109,174]. In support of this, truncation of the C-terminal 25 amino acid residues of the cytoplasmic tail of ICAM-3 disrupted ICAM-3-mediated costimulation of T cells, as shown by a reduction in IL-2 production [131]. Furthermore, the stimulation of T cells with agonistic CD3 Abs or neutrophils with FMLP, in combination with antibodies against ICAM-3, promotes Tyr phosphorylation by members of the Src family of kinases [175,176]. ICAM-3 costimulation of T lymphocytes activates/phosphorylates PLC-γ [165], whereas in neutrophils, ICAM-3 itself is phosphorylated on Tyr [177]. Regarding this, the regulation of ICAM-3-induced Tyr phosphorylation by the phosphatase activity of CD45 has been observed in LFA-1-mediated homotypic T-cell interactions [178], suggesting that CD45 may be key in controlling ICAM-3-mediated costimulation. On the other hand, ICAM-3 can also be phosphorylated on Ser, since upon ICAM-3-mediated costimulation, Ser489 is susceptible to phosphorylation by PKC-θ and regulates IL-2 production in Jurkat T cells [131]. Although ICAM-3 appears to be a well-established target of Tyr/Ser kinases, little is known about the significance of these posttranslational modifications in the context of the immune responses. 

In more physiological settings, it has been reported that thymocyte development and T-cell function are defective in ICAM-1 knockout mice. Specifically, ICAM-1-deficient Treg cells show impaired activation and control of colon inflammation in an experimental mouse model of colitis [179]. Accordingly, it has been proposed that, in the periphery, anti-inflammatory cytokines (e.g., IL-10, IL-4 and TGF-β) from ICAM-mediated homotypic interactions of activated CD4^+^ T cells may promote regulatory functions on resting CD4^+^ T cells through a mechanism dependent on LFA-1 activation in cis [180]. In CD8^+^ T lymphocytes, ICAM-1-mediated homotypic interactions also regulate effector functions since loss of ICAM-1 in these cells enhances cytotoxicity by increasing IFN-γ and granzyme B production but reduces the expression of the checkpoint inhibitor CTLA-4 [181]. It has been previously reported that during homotypic T-cell interactions, LFA-1/ICAM-1 is organized in multifocal synapses [182]. Whether the organization of LFA-1/ICAMs in the IS formed between activated and resting T cells is also multifocal and plays a role in the acquisition of the regulatory phenotype of resting T cells remains an open question.

ICAMs are also important for the fate of B cells. Two-photon microscopy studies of mouse LNs have shown that ICAM-1 and -2 on B cells help to establish cognate interactions with helper T cells at the border of B-cell–T-cell areas, forming dynamic conjugates that, under the guidance of the B cell, move towards the follicle, where ICAMs are essential for the establishment of long-lasting Ag-specific interactions between B cells and follicular T cells (Tfh), promoting the selection of B-cell clones and their expansion in germinal centres [183,184], which are essential for the generation of protective humoral immune responses.

## 5. ICAMs in Disease

The importance of the interactions between β2 integrins and ICAMs in human health was first highlighted by the study of leukocyte adhesion deficiency (LAD), a group of genetic disorders consisting of a defect in cellular adhesion molecules that results in combined B-cell and T-cell immunodeficiency [185]. Type I LAD patients express defective β2- integrin mutants, whereas β1- and β3- integrin levels are also altered in type III LAD, affecting all integrins expressed on lymphocytes as a consequence of mutations in the FERMT3 gene, leading to defective kindlin-3 expression and integrin activation [186]. Even though most of the literature and research on LAD have focused on integrins. But, given that ICAMs play important costimulatory function in lymphocyte activation and migration it is likely that ICAMs’ inability to properly contact with β2 integrins also contributes to immune defects in LAD and other diseases in which the immune response is involved.

### 5.1. Dry Eye Disease

LFA-1/ICAM-1 plays an important role in cell-mediated immune responses associated with dry eye disease (DED), also known as keratoconjunctivitis sicca, an autoimmune disease characterized by inflammation of the lacrimal gland and ocular surface [187]. The interaction between ICAM-1 and LFA-1 is thought to be important for the progression of DED, promoting T-cell activation by eye-resident APCs, migration of activated CD4^+^ T cells to the ocular surface and their retention in the conjunctival epithelium. Highlighting the importance of the LFA1/ICAM-1 interaction in this pathology, the clinical efficacy of lifitegrast, a small-molecule tetrahydroisoquinoline antagonist of LFA-1 that blocks its binding to ICAM-1 [188], has been recently demonstrated in DED [189].

### 5.2. Cardiovascular Disease 

ICAM-1-deficient mice are protected against ischemia. Complications of middle cerebral artery occlusion were reduced in ICAM-1-deficient mice compared to wild-type animals, and cerebral blood flow was increased 3.5-fold [190]. Similarly, ICAM-1-deficient mice are protected against renal ischemic injury [191]. On the other hand, the increased surface expression of ICAM-1 and other adhesion molecules is characteristic of chronic inflammation-induced endothelial dysfunction [192]. In this context, hypoglycosylated N-glycoforms of ICAM-1 bind monocytes with higher affinity than ICAM-1 glycoforms containing complex α-2,6-sialylated N-glycans [43]. Those hypoglycosylated ICAM-1 forms also regulate endothelial signalling in different mechanisms than hyperglycosylated ones do. ICAM-1 glycoforms with high-mannose (HM) or α-2,6-sialylated epitopes are present in human and mouse atherosclerotic lesions. Furthermore, HM ICAM-1 is associated with an increased presence of macrophage in these lesions, whereas the α-2,6-sialylated form is not. Regarding this, it is worth noting that both HM and α-2,6-sialylated ICAM-1 N-glycoforms are present in basilic vein samples from hemodialysis patients with arteriovenous fistula maturation failure [49].

### 5.3. Intestinal Inflammation

ICAMs also play a role in intestinal inflammation. ICAM-1 levels are increased in colon tissue from ulcerative colitis and Crohn’s disease, whereas ICAM-3 is elevated only in Crohn’s disease; in contrast, ICAM-2 levels are unaltered in both diseases [193]. It has been suggested that the recruitment of lymphocytes to the inflammatory foci in these diseases makes use of different adhesion mechanisms based on the different expressions of ICAMs. Along the same lines, in an experimental mouse model of colitis induced by dextran sodium sulphate, ICAM-1 deficiency protected mice against severe forms of the disease, with lower mortality and the absence of clinical symptoms [194]. Therefore, the study of ICAM-1 may be relevant to understanding the pathogenesis of some types of intestinal inflammatory diseases.

### 5.4. Pulmonary Fibrosis 

ICAM-1 is also involved in the inflammatory response to LPS-induced lung injury in the distal airways. In this condition, the adhesion of neutrophils to alveolar epithelial cells is significantly reduced by blocking ICAM-1 [195]. The expression levels of ICAMs are also relevant diagnostic markers of pulmonary fibrosis-related diseases (e.g., idiopathic pulmonary fibrosis and secondary interstitial fibrosis) since, in both pathologies, the levels of ICAM-1 and -2 increase in the serum of patients, with ICAM-2 being a better biomarker of severity [196], whereas the levels of ICAM-3 are not altered.

### 5.5. Autoimmunity 

In autoimmune demyelinating diseases, an ICAM-1 isoform lacking D4 in the mouse may drive pathogenesis and may be a novel therapeutic target for the treatment of multiple sclerosis (MS) [65,66]. In contrast to ICAM-1, which is a marker of inflammation given its inducible nature and expression on many cells aside from the immunocompetent cells, ICAM-3 may be a useful marker of a low level of activation in the immune system [114]. This principle has been applied to the search for markers of disease activation in cerebrospinal fluid and blood for the development of MS treatments [197]. ICAMs are also important in the pathogenesis of many skin disorders, such as cutaneous T-cell lymphoma, which shares important clinical and histological autoimmune features, playing a key role in the selective localization of T cells to the skin. Patients with this disease have high levels of sICAM-1 and sICAM-3 in the serum compared with healthy individuals or patients with inflammatory dermatosis [198,199]. sICAM-3 is also elevated in the sera of patients with psoriasis, another autoimmune dermatological disease [200]. The development and use of soluble ICAMs as markers of inflammatory disease may help to improve diagnosis.

### 5.6. Hematopoietic Stem Cell Transplantation (HSCT)

The clinical use of mesenchymal stem cell exosomes loaded with a miRNA that inhibits ICAM-1 expression (MiR-223) was able to restrain T-cell adhesion and migration in mouse lymphatic endothelial cells. The infusion of this miRNA has been shown to attenuate symptoms of acute graft-versus-host disease (aGvHD) by reducing donor T-cell infiltration into host organs and the production of proinflammatory cytokines, such as IFN-γ, TNF-α and IL-17 [201]. However, when ICAM-1 deficiency occurs in the bone marrow niche, the quiescence and repopulation of hematopoietic stem cells are impaired [202]. Therefore, ICAM-1 may play a dual role in HSCT. On the one hand, it may promote unintended inflammatory graft responses, while, on the other hand, it may play a key role in maintaining HSC quiescence and repopulation of the hematopoietic niches.

### 5.7. Cancer

ICAM-1 is upregulated in human melanoma cells during their metastasis from the primary tumour site to lymph nodes [90]. This upregulation is necessary to initiate the lymphatic spread of the disease but dispensable for metastasis to other organs. In cervical cancer, the induction of ICAM-3 has been shown to confer radioresistance to tumour cells and anticancer drug resistance via activation of the Akt/ERK-CREB-2 signalling pathway, which induces cancer cell proliferation and reduces apoptosis [172]. This signalling pathway increases the metastatic potential of non-small lung cancer cells through the induction of metalloproteases-like MMP-2 and -9 and the degradation of the extracellular matrix [203]. ICAM-3 is also involved in the metastasis of breast cancer [173]. This occurs through its binding to LFA-1 and ERM proteins on the cell lamellipodium, where they generate the tension that pulls cells apart and into motion during metastasis. In lung cancer, tertiary lymphoid structures were found adjacent to tumours and were rich with naive T cells. The gene expression signature found in these structures included adhesion molecules such as ICAM-2 and -3 [204]. This may support tumour-specific T-cell migration into these structures, where T-cell priming may occur, opening new opportunities for cancer immunotherapy.

Taken together, these studies propose ICAMs as potential biomarkers and therapeutic targets for the development of interventional therapies. Many of these experimental therapies are based on the use of ICAM blocking Abs and small molecules that interfere with the interaction between ICAMs and β2 integrins but also on the use of soluble recombinant ICAMs and ICAM-derived peptides with similar functions. Remarkably, the innovative use of ICAMs as target molecules to deliver exosomes, pharmacological drugs in polymer carriers or siRNAs against cancer and inflammatory cells [96,205,206] or to guide anti-inflammatory mesenchymal stem cells (MSCs) to inflamed tissues [207] and CAR-T cells in immunotherapy [208] has emerged as a promising strategy for the treatment of cancer and autoimmune diseases. 

## 6. Conclusions and Future Perspectives

A large body of evidence from seminal studies has long demonstrated that ICAMs are adhesion molecules by themselves, with important signalling functions in the immune system. However, some of their functions are shared with integrins in cognate interactions between lymphocytes and APCs (Figure 2B–E), fostering lymphocyte activation and proliferation. Why some of the costimulatory actions of LFA-1 and ICAMs are so similar is an open question. A tentative answer to this question may come from the well-documented cross-talk between ICAMs and the activation of β2 integrins, which has been exemplified by LFA-1 on T cells. When T cells were stimulated with a peptide from the LFA-1-binding D1 of ICAM-2, their interaction with ICAM-1, -2 and -3 increased, as did the binding of ICAM-1 to LFA-1 when cells were either bound to recombinant ICAM-3-Fc chimeric fusion protein or stimulated with agonistic anti-ICAM-3 Abs [115,209,210,211], suggesting that ICAMs are able to increase LFA-1 affinity and/or avidity, and that LFA-1-shared ICAM signalling in T cells would take place via LFA-1. Notwithstanding the above, there is clear evidence that ICAMs can also signal by themselves in a specific manner, such as the important anti-apoptotic effects of ICAM-1, -2 and 3 in lymphocytes and cancer cells. Whether ICAMs are functionally redundant or have their own specific properties and functions is another challenging question given their broad expression on leukocytes and endothelial cells. In this regard, although recent studies have questioned the function of LFA-1 in a range of T-cell responses primed by DC-mediated antigen presentation in vivo, it is well established that the selective interaction of ICAMs with this integrin plays a predominant role from the lymphocyte side [212]. It has been reported that in lymphocyte trafficking, both ICAM-1 and -2 are important for recirculation, whereas in inflamed skin and lungs, only ICAM-1 is essential for lymphocyte arrest in the tissue [213]. Moreover, ICAM-2 and, to a lesser extent, ICAM-3, but not ICAM-1, promoted perforin release via Src kinases during the interaction of CD8^+^ CD56^+^ NK cells with target cells, whereas both ICAM-2 and -3 can differentially modulate human T-cell development in vitro, fostering the production of functional memory CD4^+^ T cells [214,215]. However, how different ICAMs can induce different LFA-1 signalling is largely unknown. On the other hand, ICAMs are involved in a wide range of inflammatory diseases, from autoimmunity to cancer, and they are emerging as promising prognostic biomarkers and therapeutic targets. Further investigation leading to a better understanding of their functions and mechanisms of action in the immune response will help in the development of new therapies for the treatment of inflammatory diseases.

## Figures and Tables

**Figure 1 cells-13-00339-f001:**
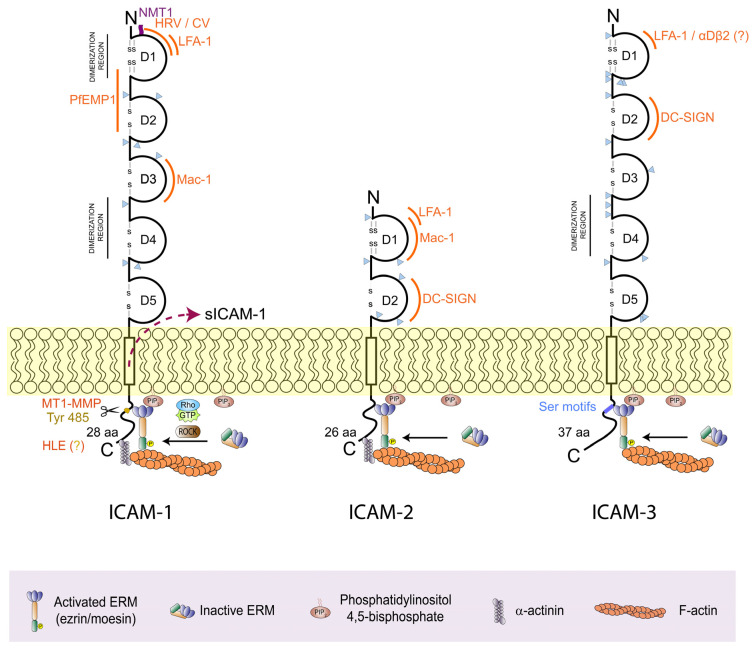
Structure, posttranslational modifications and ligands of immune ICAMs at the N-terminal (N) extracellular and C-terminal (C) intracellular domains. The binding sites of ICAMs to β2 integrins (LFA-1, Mac-1 and the putative α_D_β_2_ binding site) and DC-SIGN; human rhinoviruses (HRV) and coxsackieviruses (CV) and the *Toxoplasma gondii* protein PfEMP1 are depicted on the corresponding Ig-like domains (D1-5) of ICAM-1, -2 and -3. Dimerization regions of ICAM-1 and -3, the N-myristoylation site (NMT1) of ICAM-1, and glycosylation sites (small triangles) for all three ICAMs are also shown. The interaction of ICAM-1 and -2 with α-actinin and the interaction of ICAM-1, -2, and -3 with ERMs (ezrin and moesin) activated by PIP2 binding and RhoGTPase (Rho-GTP)-Rho-associated protein kinase (ROCK)-mediated phosphorylation are depicted near their cytoplasmic tails and amino-acid lengths. The cytoplasmic tails of ICAM-1 and ICAM-3 show the proteolytic enzymes and recognition site (MT1-MMP/Tyr485 and human leukocyte elastase (HLE)/unknown recognition site (?)) involved in ICAM-1 cleavage/soluble (s)ICAM-1 release and the Ser motifs that regulate ERM binding, respectively.

**Figure 2 cells-13-00339-f002:**
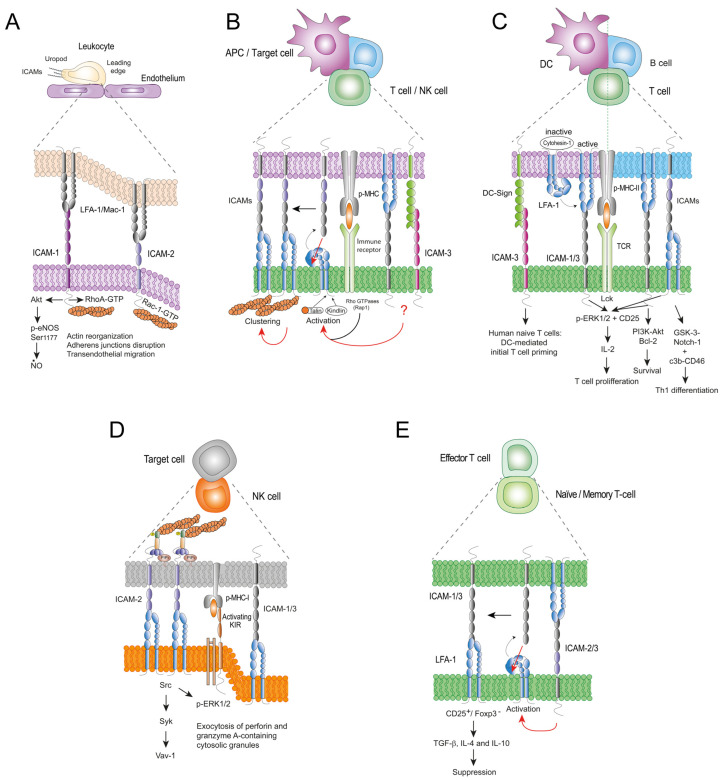
ICAMs-mediated cell-cell interactions in immunity. (**A**) ICAM-1 and ICAM-2 differentially activate actin cytoskeleton-associated Rho GTPases (RhoA and Rac-1) and signalling during leukocyte adhesion and migration on endothelial cells. (**B**) Regulation of LFA-1 activation by ICAMs during cognate interactions in the immune system. In effector immune cells, ICAMs may regulate LFA-1 at three levels (red arrows): (i) in “cis” through unknown (?) mechanisms, during their binding to LFA-1 and DC-SIGN; (ii) in “trans” by direct binding to the ICAM-binding site of partially activated LFA-1 and; (iii) fostering F-actin rearrangements and LFA-1 clustering. (**C**) TCR-associated ICAM-1, -3 and LFA-1 costimulatory signalling in T cells during cognate interactions with dendritic (DC, magenta) or B cells (blue). Cytohesin-1-inactivated LFA-1 and DC-SIGN are represented as characteristic of DCs. Shared and non-shared ICAM-1, -3 and LFA-1 costimulatory signalling and functions are depicted. (**D**) NK cell receptor-associated LFA-1 costimulatory signalling induced by ERM-mediated ICAM-2 clustering and binding during MHC-I-mediated interactions between NK cells and target cells. LFA-1 binding to ICAM-1 and -3 are also represented as less efficient inducers of LFA-1 costimulation. (**E**) Regulatory phenotype induced in naïve T cells during ICAMs-LFA-1-mediated homotypic interactions with activated T cells. “Cis” activation and regulatory signalling of LFA-1 in naïve T cells by the binding of ICAM-2 and -3 to LFA-1 on activated effector T cells are represented.

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
