# Peer review of "ICAMs in Immunity, Intercellular Adhesion and Communication"

_cells, 2024, doi:10.3390/cells13040339_

Round 1
Reviewer 1 Report
Comments and Suggestions for Authors
This is a well written review on ICAMs by a well established group. I have only a few comments. Relatively recent work has shown that integrins may bind in cis to ICAM-1 ( Fan et al. Cell Reports 26,119) and further references. These results are quite interesting and could be added.
Minor points: line 76, alfaXbeta2 and alfaDbeta2 are primarily expressed on monocytes/macrophages. Line 505, in LADI, beta2 integrins must not be lacking, they can be mutated.
Author Response
Reviewer #1
-This is a well written review on ICAMs by a well established group. I have only a few comments. Relatively recent work has shown that integrins may bind in cis to ICAM-1 ( Fan et al. Cell Reports 26,119) and further references. These results are quite interesting and could be added.
Reply:
We very much appreciate the constructive reviewer’s comments which have been taken into account to improve the manuscript. In the new version of the manuscript, we have included the interesting recent works by Fan et al. regarding the interaction between ICAMs and b2 integrins in cis (Page 6: lines 185-194 and new references 59 and 60).
-Minor points: line 76, alfaXbeta2 and alfaDbeta2 are primarily expressed on monocytes/macrophages. Line 505, in LADI, beta2 integrins must not be lacking, they can be mutated.
Reply:
We have specified that integrins aXb2 and aDb2 are mainly expressed on monocytes and macrophages (Page 2: lines 75-77) and that type I LAD patients express mutant forms of b2 integrins (Page 15: lines 517-518).
New references:
(59) Fan Z, Kiosses WB, Sun H, Orecchioni M, Ghosheh Y, Zajonc DM, Arnaout MA, Gutierrez E, Groisman A, Ginsberg MH et al: High-Affinity Bent beta(2)-Integrin Molecules in Arresting Neutrophils Face Each Other through Binding to ICAMs In cis. Cell Rep 2019, 26(1):119-130 e115.
(60) Fan Z, McArdle S, Marki A, Mikulski Z, Gutierrez E, Engelhardt B, Deutsch U, Ginsberg M, Groisman A, Ley K: Neutrophil recruitment limited by high-affinity bent beta2 integrin binding ligand in cis. Nat Commun 2016, 7:12658.
Reviewer 2 Report
Comments and Suggestions for Authors
This is an excellent review that covers both the basic biology of ICAMs as well as provides insight into the pathology of diseases related to ICAM dysfunction. The review is well-written and organized in a logical fashion, with excellent illustrations. This review will be a welcome addition to the literature.
Author Response
Reviewer #2
- This is an excellent review that covers both the basic biology of ICAMs as well as provides insight into the pathology of diseases related to ICAM dysfunction. The review is well-written and organized in a logical fashion, with excellent illustrations. This review will be a welcome addition to the literature.
We thank the reviewer for her/his positive comments on the review.
Reviewer 3 Report
Comments and Suggestions for Authors
To Author:
Intercellular adhesion molecules (ICAMs) play an important role in regulating cellular interactions. In this review, the authors summarized the functions of ICAMs and related signaling pathways in detail and discusses in detail the role of ICAMs in related diseases and related molecular mechanisms. I considered this review paper to be significant. However, I have several suggestions before it can be accepted.
Comments:
(1) References are missing in some places in the text. Such as line 64 to 65.
(2) When the author introduced the glycosylation modification of ICAMs, he only introduced the N-glycosylation and did not introduce the O-glycosylation of ICAMs.
(3) While introducing the role of ICAMs in related diseases, the authors should also discuss the therapeutic significance of targeting ICAMs molecules in related diseases.
Author Response
Reviewer #3
- Intercellular adhesion molecules (ICAMs) play an important role in regulating cellular interactions. In this review, the authors summarized the functions of ICAMs and related signaling pathways in detail and discussed in detail the role of ICAMs in related diseases and related molecular mechanisms. I considered this review paper to be significant. However, I have several suggestions before it can be accepted.
Reply:
We appreciate the positive comments of the reviewer, as well as his/her constructive suggestions, that have helped us to improve the manuscript.
- Comments:
(1) References are missing in some places in the text. Such as line 64 to 65.
(2) When the author introduced the glycosylation modification of ICAMs, he only introduced the N-glycosylation and did not introduce the O-glycosylation of ICAMs.
(3) While introducing the role of ICAMs in related diseases, the authors should also discuss the therapeutic significance of targeting ICAMs molecules in related diseases.
Reply to Comments:
1.- New references 7, 8, 42, 59, 60, 93, 206, 207, 208 and 209 have been added in the revised form of the manuscript.
2.- To our knowledge, and after an exhaustive bibliographic search, ICAMs are mainly glycosylated on Asn. O-glycosylation can be detected by LC-MS if the N-linked glycans are first removed using PNGaseF. However, it has been reported that consensus sites for O-glycosylation are not present in ICAM-3 and the unusually high frequency of N-glycans may account for resistance to endo-glycosidases, especially in ICAM-1 and ICAM-2, whose glycosylations appear to be large, highly branched or extended complex-type structures (Hughes, New comprehensive Biochemistry (1997) 29, Part B, Pages 507-570, https://doi.org/10.1016/S0167-7306(08)60627-4). In addition, despite the presence of seven predicted O-glycosylation sites on ICAM-1, a detailed carbohydrate analysis of its five soluble glycoforms expressed in CHO cells, demonstrated that the most abundant monosaccharides present on ICAM-1 are N-acetylglucosamine, galactose and mannose, confirming the predominance of complex-type N-glycans (Otto et al. 2004 J. Biol. Chem. 279: 35201-35209). This new reference has been added to the resubmitted manuscript and commented on (Page 4: lines 133-136, Reference 42).
3.- Following the helpful reviewer’s suggestion, a paragraph discussing the possible therapeutic significance of targeting ICAMs in inflammatory diseases has been added at the end of the manuscript’s “ICAMs in disease” section (Page 17: lines 615-624).
New references:
(7) Fukuda M, Hiraoka N, Yeh JC: C-type lectins and sialyl Lewis X oligosaccharides. Versatile roles in cell-cell interaction. J Cell Biol 1999, 147(3):467-470.
(8) Ivetic A, Hoskins Green HL, Hart SJ: L-selectin: A Major Regulator of Leukocyte Adhesion, Migration and Signaling. Front Immunol 2019, 10:1068.
(42) Otto VI, Schurpf T, Folkers G, Cummings RD: Sialylated complex-type N-glycans enhance the signaling activity of soluble intercellular adhesion molecule-1 in mouse astrocytes. J Biol Chem 2004, 279(34):35201-35209.
(59) Fan Z, Kiosses WB, Sun H, Orecchioni M, Ghosheh Y, Zajonc DM, Arnaout MA, Gutierrez E, Groisman A, Ginsberg MH et al: High-Affinity Bent beta(2)-Integrin Molecules in Arresting Neutrophils Face Each Other through Binding to ICAMs In cis. Cell Rep 2019, 26(1):119-130 e115.
(60) Fan Z, McArdle S, Marki A, Mikulski Z, Gutierrez E, Engelhardt B, Deutsch U, Ginsberg M, Groisman A, Ley K: Neutrophil recruitment limited by high-affinity bent beta2 integrin binding ligand in cis. Nat Commun 2016, 7:12658.
(93) Lee HM, Choi EJ, Kim JH, Kim TD, Kim YK, Kang C, Gho YS: A membranous form of ICAM-1 on exosomes efficiently blocks leukocyte adhesion to activated endothelial cells. Biochemical and biophysical research communications 2010, 397(2):251-256.
(206) Muro S, Garnacho C, Champion JA, Leferovich J, Gajewski C, Schuchman EH, Mitragotri S, Muzykantov VR: Control of endothelial targeting and intracellular delivery of therapeutic enzymes by modulating the size and shape of ICAM-1-targeted carriers. Mol Ther 2008, 16(8):1450-1458.
(207) Wang X, Liu W, Rui Z, Zheng W, Tan J, Li N, Yu Y: Immunotherapy with a biologically active ICAM-1 mAb and an siRNA targeting TSHR in a BALB/c mouse model of Graves' disease. Endokrynol Pol 2021, 72(6):592-600.
(208) Zheng S, Huang K, Xia W, Shi J, Liu Q, Zhang X, Li G, Chen J, Wang T, Chen X et al: Mesenchymal Stromal Cells Rapidly Suppress TCR Signaling-Mediated Cytokine Transcription in Activated T Cells Through the ICAM-1/CD43 Interaction. Front Immunol 2021, 12:609544.
(209) Min IM, Shevlin E, Vedvyas Y, Zaman M, Wyrwas B, Scognamiglio T, Moore MD, Wang W, Park S, Park S et al: CAR T Therapy Targeting ICAM-1 Eliminates Advanced Human Thyroid Tumors. Clin Cancer Res 2017, 23(24):7569-7583.
Reviewer 4 Report
Comments and Suggestions for Authors
In this review, the authors cited a large amount of data to illustrate the important role of ICAM in signal transduction, and sorted out the important functions in the cancer and APCs, especially in the activation, differentiation and migration of T calls. This study has certain clinical significance and provides a lot of information. Overall, this article is informative and should be published without any revisions.

Author Response
Reviewer #4
In this review, the authors cited a large amount of data to illustrate the important role of ICAM in signal transduction, and sorted out the important functions in the cancer and APCs, especially in the activation, differentiation and migration of T cells. This study has certain clinical significance and provides a lot of information. Overall, this article is informative and should be published without any revisions.
We thank the reviewer and appreciate very much her/his positive comments on the review.